# Single Nucleotide Polymorphisms in *XMN1-HBG2*, *HBS1L*-*MYB*, and *BCL11A* and Their Relation to High Fetal Hemoglobin Levels That Alleviate Anemia

**DOI:** 10.3390/diagnostics12061374

**Published:** 2022-06-02

**Authors:** Siti Nur Nabeela A’ifah Mohammad, Salfarina Iberahim, Wan Suriana Wan Ab Rahman, Mohd Nazri Hassan, Hisham Atan Edinur, Maryam Azlan, Zefarina Zulkafli

**Affiliations:** 1Department of Haematology, School of Medical Sciences, Health Campus, Universiti Sains Malaysia, Kubang Kerian 16150, Kelantan, Malaysia; aifahnabeela@gmail.com (S.N.N.A.M.); salfarina@usm.my (S.I.); nazrihas@usm.my (M.N.H.); 2Transfusion Medicine Unit, Hospital Universiti Sains Malaysia, Kubang Kerian 16150, Kelantan, Malaysia; suriana@usm.my; 3School of Dental Sciences, Health Campus, Universiti Sains Malaysia, Kubang Kerian 16150, Kelantan, Malaysia; 4School of Health Sciences, Health Campus, Universiti Sains Malaysia, Kubang Kerian 16150, Kelantan, Malaysia; edinur@usm.my (H.A.E.); maryamazlan@usm.my (M.A.)

**Keywords:** fetal hemoglobin, anemia, single nucleotide polymorphism, *XMN1*-*HBG2*, *HBS1L*-MYB, *BCL11A*

## Abstract

Anemia is a condition in which red blood cells and/or hemoglobin (Hb) concentrations are decreased below the normal range, resulting in a lack of oxygen being transported to tissues and organs. Those afflicted with this condition may feel lethargic and weak, which reduces their quality of life. The condition may be manifested in inherited blood disorders, such as thalassemia and sickle cell disease, whereas acquired disorders include aplastic anemia, chronic disease, drug toxicity, pregnancy, and nutritional deficiency. The augmentation of fetal hemoglobin (HbF) results in the reduction in clinical symptoms in beta-hemoglobinopathies. Several transcription factors as well as medications such as hydroxyurea may help red blood cells produce more HbF. HbF expression increases with the downregulation of three main quantitative trait loci, namely, the *XMN1-HBG2*, *HBS1L*-MYB, and *BCL11A* genes. These genes contain single nucleotide polymorphisms (SNPs) that modulate the expression of HbF differently in various populations. Allele discrimination is important in SNP genotyping and is widely applied in many assays. In conclusion, the expression of HbF with a genetic modifier is crucial in determining the severity of anemic diseases, and genetic modification of HbF expression may offer clinical benefits in diagnosis and disease management.

## 1. Introduction

Anemia is a common hematological condition in acquired and inherited disorders, with diverse pathophysiology, such as infections, iron deficiency, kidney disease, cancer, thalassemia, sickle cell disease (SCD), and hereditary persistence fetal hemoglobin (HPFH) [1,2,3]. Anemia is characterized by a decrease in normal hemoglobin concentration and, besides a reticulocyte count, is also classified according to mean corpuscular volume (MCV) due to altered red blood cell morphology that leads to microcytic, normocytic, and macrocytic anemias [1,4,5].

Increased HbF levels due to reactivation or overexpression of the *HBG* gene and subsequent production of γ-globin alleviate the severity of anemia, especially in hemoglobinopathies, such as SCD and thalassemia [6,7]. The risk of mortality and morbidity in those diseases are decreased in patients who have high levels of HbF [7]. High HbF levels can be due to the effects of drugs, autoimmune disease, pregnancy, malignancy, diabetes, genetic modifiers, and other factors [8].

Genetic modifiers are known as single nucleotide polymorphisms (SNPs) [9,10,11]. SNPs are the most common type of human genetic variation, as shown in Figure 1. There are two types of SNPs: synonymous SNPs and nonsynonymous SNPs. Synonymous SNPs are known as silent SNPs due to unchanged amino acid, whereas nonsynonymous SNPs may produce disease by affecting amino acids [12]. Each SNP consists of a difference in a single DNA building block, called a nucleotide. For example, an SNP may replace the nucleotide alanine (A) with the nucleotide guanine (G) in a stretch of DNA. The polymorphism can occur by mutation, resulting in nucleotide changes, such as insertion, deletion, or rearrangement of a nucleotide. Therefore, SNPs are among the genetic polymorphisms that consist of a difference in DNA sequence among individuals, groups, or populations, with the production of altered proteins [13]. SNPs have advantages, which can provide genetic information such as ethnicity, physical traits, or phenotype and can be used as a genetic marker in the human genome for genetic mapping or studies of associations between polymorphism and disease or drug response [14,15,16,17]. The distribution of SNPs throughout genes has been an important biomarker in many applications, including quantitative trait loci mapping, pedigree analysis, and population genetics [18]. Moreover, the presence of SNPs may provide advantages in clinical diagnosis and disease management, such as for patients with blood disorders, who may be offered potential targeted therapies. However, SNPs also have limitations or disadvantages that affect the routine approach, such as the complement probabilities or discrimination power when more SNPs are present in assays. Yet, more loci and amplification products of SNPs will help make data clarification more strenuous and precise [16].

## 2. Fetal Hemoglobin

HbF is produced in a fetus at around six weeks of pregnancy and constitutes 98% of its total hemoglobin content until birth [19]. HbF consists of a heterogenous structure (2 α and 2 γ subunits), with two types of γ-globulin chains made of glycine and alanine. HbF has stronger oxygen binding than does adult hemoglobin (HbA), thus enabling the unborn child to draw oxygen from the mother’s blood. The switching from HbF to HbA occurs soon after birth. The HbF level declines to around five per cent of the total hemoglobin level in normal 6-month-old babies, and by the time they reach 2 years old, the level will have dropped to less than one per cent, where it remains throughout life.

HbA will become the main hemoglobin produced by the body, at 95% to 98% of the total Hb, followed by HbA2 (2.5% to 3.5%) [7]. The concentration of HbF is determined by the production of the γ-globin chain, and the rise in HbF levels may ameliorate the severity of anemia in adults with blood disorders [20]. In adults, alterations in the *HBB* gene that produces the β-globin chain or point mutations upstream of the *HBG* gene (which produces the γ-globin chain) have been linked to unusually high γ-globin chain expression. The reactivation of HbF production may occur in certain acquired and inherited conditions. For example, alterations in the β-globin chain structure in SCD, β-thalassemia, δβ-thalassemia, and HPFH may lead to increased HbF production in red blood cells as the γ-globin chain is expressed as a compensatory mechanism [19,21,22].

HPFH is a heterogenous condition that has high expression of γ-globin chains caused by a point mutation in the promoter region of the gene or an imbalance of the α/β-globin chain as a result of a deletion in the β-globin gene [23,24,25]. HPFH patients have around 10% to 40% HbF of their total hemoglobin, but in those with a genetic deletion in other globin genes, the value may reach 100% [25,26]. HbF is also elevated in pregnant women, cancer patients being treated with hydroxyurea, and in patients with aplastic anemia, hepatoma, or myeloid leukemia [25,26,27].

Both SCD and thalassemia patients who have high HbF levels have therapeutic advantages, including reduced disease severity and better outcomes [7]. Therapeutic agents, such as the inducer drugs hydroxyurea and thalidomide, are widely used in treating SCD, thalassemia, myelodyplastic anemia, sideroblast anemia, and leukemia by increasing patients’ HbF values [26,28,29,30,31,32]. Hydroxyurea is metabolized to produce nitric oxide and activates gene transcription to promote HbF synthesis [26]. Ribonucleotide reductase is inhibited, thus blocking DNA synthesis, leading to cellular cytotoxicity and suppression of erythroid progenitors, which affects the erythropoiesis kinetics and physiology and leads to recruitment of erythroid progenitors with increased HbF levels [26,28,30,32,33]. A once-daily treatment of SCD with hydroxyurea may produce intermittent cytotoxic suppression of erythroid progenitors and reduce cell-stress signaling, thus altering the kinetics of erythropoiesis and elevated HbF values with erythroid progenitor enrollment (33).

Patients who are unresponsive to hydroxyurea treatment can be treated with sodium butyrate containing micro RNAs to induce HbF synthesis [34,35]. Thalidomide is a synthetic glutamic acid derivative used to treat autoimmune diseases, hematologic disorders, and multiple myeloma by increasing HbF levels and thereby alleviating symptoms [36]. Thalidomide also has a benign toxicity profile and remarkable efficacy in reducing the severity of anemia [37]. The increase in HbF synthesis in patients with blood disorders refines tissue oxygenation and reduces inflammation besides reducing red blood cell membrane damage, diminishing HbS polymerization, and decreasing vasoconstriction and thrombosis [29]. The use of thalidomide may also reduce the incidence of hospitalization and blood transfusion [30].

## 3. Polymorphisms Regulate the Expression of Fetal Hemoglobin

The expression of HbF is facilitated by the presence of SNPs at three loci: *XMN1-HBG2* at 11p15.4, *BCL11A* at 2p16.1, and the *HBS1L-MYB* intergenic region at 6q23.3, as shown in Figure 2. These three loci are aligned to increase HbF, accounting for 20–50% of inter-individual variation in HbF values [7]. The increase in HbF ameliorates the severity of anemia, which can decrease the need for transfusion and the severity of renal impairment. Regulation of the HbF value requires precise communication among transcription factors to achieve downregulation or activation of genes in these loci (Figure 2).

The presence of SNPs may have implications in a disease, such as increased HbF levels in common and complex diseases or conditions, e.g., drug-induced conditions, SCD, β-thalassemia, anemia, diabetes, chronic kidney failure, malignancies, or obesity in many regions of the world, such as Europe, the Middle East, and Southeast Asia, including Malaysia [20,38,39,40,41,42,43,44]. Several SNPs, such as rs11886868, rs766432, rs9399137, rs742144, rs9399137, rs28384513, and rs4671393, are associated with erythropoiesis and HbF augmentation (Table 1 and Figure 2) [45,46,47]. SNPs are one of genetic variations that may alter the outcome of downstream processes by influencing the promoter activity in gene expression and stabilizing mRNA, thus allowing expression of a disease or phenotype and traits [12].

### 3.1. Epidemiology and Molecular Aspects of XMN1-HBG2

*XMN1-HBG2* is a genetic modifier in β-thalassemia, located at −158 upstream of the γ-globin gene on chromosome 11. The polymorphism of *XMN1* is expressed in the γ-globin gene in the promoter region by nucleotide substitution of C→T (rs7482144), thus increasing HbF levels and ameliorating the phenotype of β-thalassemia and anemia (Figure 2 [25,42]. Because of increased HbF levels, patients with heterozygous *XMN1*-*HBG2* have only modest clinical manifestations [41].

According to studies on human erythroid progenitor cells, hydroxyurea treatment may increase erythroid differentiation and HbF levels by inducing the expression of the γ-globin gene [64]. While most SNPs in *XMN1-HBG2* polymorphism are associated with high HbF values, HbF susceptibility to hydroxyurea treatment is regulated by alternative pathways that protect hematopoietic cells from stress-induced death and push them to endpoint erythroid differentiation [64]. The presence of *XMN1-HBG2* polymorphism and treatment interventions, such as with azacytidine and hydroxyurea agents, are associated with minimal need for blood transfusion among blood disorder patients by increasing their HbF levels, resulting in less severe disease.

In Malaysia, both genotypes (CT and TT) of *XMN1-HBG2* polymorphism are highly present in Malay β-thalassemia major individuals compared with that in Chinese patients, resulting in high HbF production in the latter [65]. Based on a study in Pakistan [66], the level of HbF is significantly increased in patients with intermediate thalassemia (around 8%) compared with that in thalassemia major patients, with the T allele in *XMN1-HBG2* frequently present in the intermediate group. The association of homozygous and heterozygous T alleles in *XMN1-HBG2* is also present in milder forms of SCD, where the HbF level is higher in TT allele carriers, followed by CT and CC genotypes (*p* < 0.01) [20]. The SNP rs7482144 in *XMN1-HBG2* has caused a 2.2% variation in HbF expression in African American SCD patients in the United States [47]. In Indonesia, the HbF expression is slightly higher, at 3.2%, and it is even higher in Syria, at 16.3% [39,42,59].

### 3.2. Epidemiology and Molecular Aspects of HBS1L-MYB

SNPs linked to hematopoietic traits may be identified in a 126-kb intergenic region between *HBS1L* (Hsp70 subfamily B suppressor 1-like gene) and the *MYB* (*MYB* proto-oncogene) at chromosome 6q23 [67,68]. *HBS1L-MYB* is an intergenic region that increases HbF values in thalassemia and sickle cell anemia and thus ameliorates anemia [67,69].

The *c-MYB* protein encoded by *MYB* has cis-regulatory and trans-acting functions in regulating hematopoiesis and erythropoiesis, thus influencing HbF values through an unidentified mechanism. The c-MYB protein is necessary for maintaining erythroid cellular proliferation and equivalence of differentiation. In vitro study of CD34 cells cultured in the presence of erythropoietin, stem cell factor, and transforming growth factor resulted in a decline in *cMYB* levels in treated cells, which contributed to an increase in γ-globin expression that ultimately increased HbF values [70]. Moreover, the protein produced by *HBS1L* is a GTPase family for elongation and may have a role in the control of cellular processes, such as protein synthesis, cytoskeleton assembly, protein trafficking, and signal transduction. Increased γ-globin expression has been induced by secretion-associated and RAS-related proteins and hydroxyurea treatment [71]. *HBS1L* and *MYB* transcription is necessary for the enhancement of HbF levels.

The *HBS1L-MYB* intergenic region consists of *HMIP-1* and *HMIP-2* genes, which are *HMIP-2A* at −84 of the *MYB* core enhancer, while *HMIP-2B* is between −71 and −63 of the *MYB* enhancer element. SNPs correlate with unusually high HbF levels based on genetic variation. The *HBS1L-MYB* intergenic polymorphism block 2 (*HMIP-2*) reveals SNPs that are strongly associated with each other, with a length of around 24 kb, such as rs28385413, rs66650371, rs9399137, rs9389269, rs9402686, rs9494145, and rs9483788 [72]. *HBS1L-MYB* is also a main regulator of erythropoiesis, and its expression may improve anemia in patients who have renal failure with the presence of rs7776054 and rs6650371 SNPs [73]. Both SNPs are significantly associated with high HbF levels that may ameliorate SCD and anemia in pregnant women [50,74].

SNPs rs4895441 (A-G), rs9399137 (T-C), and rs28384513 (A-C) are related, with 3.8% to 32% of HbF level variation in *HBS1L-MYB* among β-thalassemia/HbE and homozygous HbE in the Thai population [51]. The HbF level among SCD in patients of European descent is similar to the range in Thai patients [51,59]. The SNPs of rs48954419 (A-G), rs9399137 (T-C), and rs28384513 (A-C) have a high possibility of ameliorating severe anemia and hemoglobinopathies, especially with SNP rs9399137, due to significantly different MCV (*p* < 0.05), MCH (*p* < 0.05), and HbF levels (*p* < 0.05) [51]. Another study in a cohort from Sabah showed that rs9399137 (T-C) and rs11759553 (A-T) SNPs were frequently present in β^0^-thalassemia patients of Filipino descent (*n* = 34, MAF = 0.15 and 0.18); these patients have mean HbF values between 1.7% and 3.4% [63]. Downregulation of *MYB* gene expression and activation of *HBG* expression by termination of the orphan nuclear receptor TR2/TR4 may contribute to the regulation of HbF gene expression in SNPs of *HBS1L-MYB* [68].

### 3.3. Epidemiology and Molecular Aspect of BCL11A

The B-cell lymphoma/leukemia 11A gene (*BCL11A*), located at chromosome 2, is a transcriptional repressor of the hematopoietic system, especially in HbF silencing and switching [60]. Increased HbF levels may be present with the downregulation of *BCL11A,* which then allows the expression of γ-globin [75,76].

The remarkable hemoglobin shifting is influenced by the expression of the *KLF1* (erythroid Kruppel-like factor) gene, which stimulates β-globin production and indirectly represses γ-globin expression [77]. However, downregulation of *KLF1* in vivo has revealed a decrease in *BCL11A* expression in adult erythroid progenitors, which leads to activation of HbF production in anemic β-thalassemia, SCD, and HPFH patients [78,79]. Furthermore, *BCL11A* is expressed only when the expression of γ-globin is silenced. SOX6 (Sry-related high-mobility group HMG-box transcription) is a transcription factor that is required for the suppression of γ-globin in humans. SOX6 coexists with the *BCL11A* gene to produce high levels of HbF (up to 45% of the red blood cells), while an inconsistent knockdown of *BCL11A* or *SOX6* caused slightly lower induction of HbF production [55].

Many SNPs may be present in *BCL11A.* A few of them are associated with lower HbF levels, such as rs4671393, rs4127407, and rs7606173. However, the same SNPs have also caused high HbF values by microdeletion [80]. The role of *BCL11A* should be scrutinized further since, according to a prior study [51], the SNPs rs1426407, rs1018987, and rs11886868 did not have significant differences in hematological parameters. However, attributed to the formation of *BCL11A*, HbF levels remained greater (3.9% to 34.2%) than normal (1%) with mild or moderate anemia in homozygous HbE and β-thalassemia/HbE individuals [51].

In SCD patients, the rs11886868 (C-T) and rs4671393 (A-G) SNPs are significantly associated with high HbF levels. The switch to T and G base pairs is high in those with <15% HbF, while C and A are present in those with HbF levels >15% [52]. A previous study revealed that the abolishment of the *BCL11A* binding domain within the silencing mode at the 5′ δ-globin gene will enhance the HbF level, and this could minimize the complications of SCD [81]. Another study discovered that the rs4671393 has potential to ameliorate the anemia due to the presence of a G allele in the genotypes that can increase the HbF levels in SCD patients [82]. A study in the Sub-Saharan region found that the BCL11A loci is crucial in HbF regulation and also showed a negative association between the incidence of ischemic stroke and the level of HbF, where SCD patients with stroke had significantly low mean Hb F compared to the group that did not have stroke (*p* = 0.024) [53].

The SNPs rs6545816 (A/C), rs6545817 (A/G), rs766432 (A/C), and rs6729815 (A/G) are associated with high HbF levels (*p* < 0.001), whereas 3.3% of HbF level variation has the highest association with rs6729815 in β0-thalassemia/HbE among Thai patients [83]. Due to the stress of erythropoiesis, rs118868686 with C alleles is more strongly associated with β-thalassemia, HPFH, and SCD than in non-anemic populations in progressing to a milder phenotype [58]. A similar study of Indonesian individuals with HbE/β-thalassemia and moderate anemia found a significant association between HbF levels and SNPs rs766432 (*n* = 117) and rs11886868 (*n* = 117) through the presence of C alleles in ameliorating clinical severity [41,83]. A few SNPs, such as rs1886868, rs4671393, and rs1427407, are involved in the downregulation of *BCL11A* with elevated HbF levels in hydroxyurea treatment [57,58]; this finding indicates that the downregulation of *BCL11A* expression may be beneficial by ameliorating symptoms of anemia through increased HbF levels. Therefore, BCL11A down-regulation or domain blockage can be a guide for new targeted therapies in SCD and β-thalassemia major patients [57,58,83].

## 4. Genotyping Technologies in Single Nucleotide Polymorphism Recognition

Alteration of a single base, resulting in SNPs, may involve the insertion, substitution, fusion, or duplication of a nucleotide. These SNPs may serve as genetic markers in clinical practice, and the SNPs or mutations may be linked to disease susceptibility, etiology, and pharmacogenomics. Such genotyping has been widely accepted, especially in pharmacogenomic studies, to avoid clinical problems such as adverse drug reactions in chronic and inherited diseases [84]. The detection of SNPs also is used to identify specific sites in genes associated with high HbF levels, which may confer therapeutic benefits on patients with blood disorders [85]. Several methods use allele discrimination procedures, as shown in Figure 3 [86].

The allelic discrimination assay is a high throughput final assay for pinpointing polymorphisms in a single nucleic acid sequence. Even though each reaction involves two primer/probe pairs, genotyping the two potential polymorphisms at the SNP location in the target template sequence is necessary [86]. The primer extension consists of three types—common primer extension, specific primer extension, and allele-specific primer extension—that may straightforwardly accomplish allelic discrimination in a DNA template and are acceptable for the detection of multiple SNPs [84]. The primer extension reaction products can be detected by electrophoresis, fluorescence, pyrosequencing, mass spectrometry, or microarray [84,87].

Allele-specific probes comprising fluorescein-labeled complementary DNA or RNA can be used to differentiate SNPs through their binding stability on target sequences. Unmatched strands of the probes will be washed away, while matched pairs will bind to the SNP targets [84,87]. Hybridization-based SNP genotyping can be carried out using microarray or chip-based technologies, TaqMan assay, a combination of matrix-assisted laser desorption ionization, time-of-flight mass spectrometry quantum beads, and dynamic allele-specific hybridization [84,87,88,89]. The hybridization of two oligonucleotides occurs through ligation catalyzed by DNA ligase to form single oligonucleotides between the allele-specific probe and common oligonucleotides [90].

The ligation required to form single oligonucleotides from hybridization of two oligonucleotides requires three types of probes (two allele-specific and one common probe) for attachment of template adjacent to the SNP, which reveals a specific SNP position [87]. The ligation also uses combinatorial fluorescence energy transfer (CFET) or padlock as a specific probe. CFET consists of fluorescence that can be used with biotinylated (common probes) to detect SNP by capillary array electrophoresis [84]. In the Padlock technique, a linear oligonucleotide probe with the imitations of allele-specific and common probes is prepared for ligation at the SNP site [84,91]. The complete match forms the circular strand product by using rolling circle amplification, conventional PCR, or microarray [91].

For an enzymatic cleavage, such as flap endonuclease, a restriction enzyme is needed for discrimination of alleles in SNP genotyping [84]. The enzymatic cleavage technique consists of two molecular assays for genetic variation: restriction fragment length polymorphism (RFLP) and an Invader^®^ assay. Due to its relatively low cost and small amount of probe needed, RFLP is widely used to identify the cleavage site in double-stranded DNA for synthesizing short DNA fragments in SNP genotyping; straightforward techniques and fewer probes are needed, but the number of SNP that can be detected is limited [14,92]. The Invader^®^ assay consists of a structure-specific cleavage by flap endonuclease enzyme and probes to hybridize with the DNA target [93]. Three probes bind to the compatible 5′-site of the SNP sites, using allele-specific probes, whereas the probes are complementary to the 3′-site [14,84,94]. Thus, the three-dimensional structure is formed, and the SNP site can be directly observed by fluorescence analysis [93].

Genotyping techniques are important in biomedical applications, where they can discriminate more than one allele of SNPs. Allele discrimination methods are used in numerous well-established SNPs genotyping technologies: Beadchip, Genechip, pyrosequencing, RFLP, MassEXTEND, Taqman, ARMS-PCR, and Invader^®^. Genotyping is useful also in the detection of genetic diseases; chronic disease such as anemia, cancer, Alzheimer’s disease, arthritis, diabetes, and blood disease; and in pharmacogenomic studies. The polymorphism advanced assay methods are accurate, sensitive, error-free, and cost-effective. They are beneficial in molecular studies, especially in the healthcare industry and advanced biology [84].

## 5. Conclusions

Anemia can be a manifestation of hemoglobinopathies, such as thalassemia and SCD, that disrupt the function of red blood cells, resulting in a lack of oxygen transported to tissues and organs. The severity of anemia can be ameliorated by the production of HbF. A high HbF level may play an important role in regulating erythropoiesis in acquired or inherited conditions of anemia. The potential to increase HbF levels is also associated with the presence of SNPs within three QTLs, *XMN1-HBG2*, *HBS1L-MYB*, and *BCL11A*, which can increase the production of HbF as a compensatory mechanism in blood diseases. Many SNP genotyping technologies with allele discrimination may be used to conduct genome-wide association studies, which can provide useful information on healthcare and complications of care. Furthermore, these genetic modifiers are also associated with disease prognosis. The use of QTL in patients with high HbF values can aid in genetic counseling for inherited anemia diseases and in the prescribing of medications.

## Figures and Tables

**Figure 1 diagnostics-12-01374-f001:**
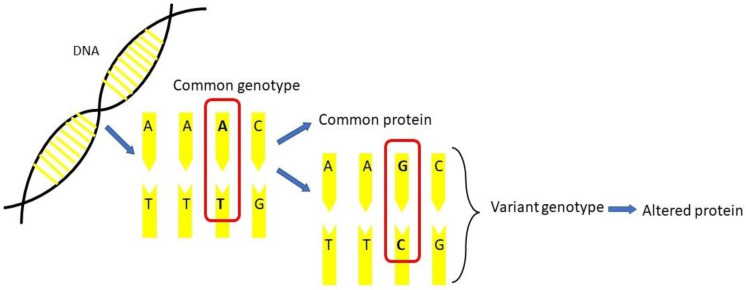
Single nucleotide polymorphisms (SNPs) in the common genotype and variant genotype of DNA (adapted from Iglesias et al., 2020).

**Figure 2 diagnostics-12-01374-f002:**
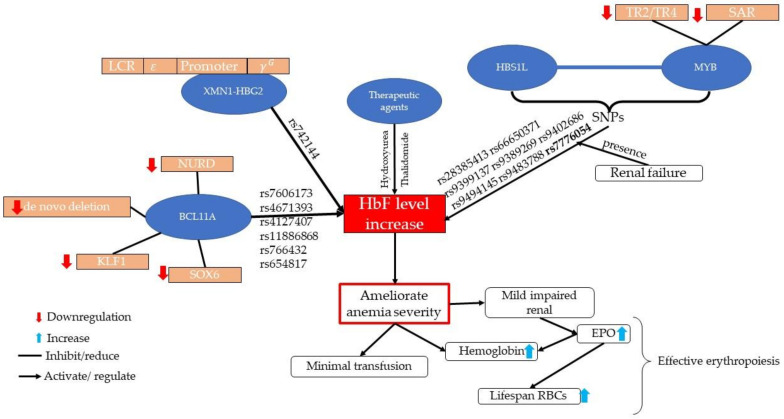
Pathway for regulation of γ-globin, possibly an increasing HbF level, and amelioration of anemia in *XMN1-HBG2*, *BCL11A*, and *HBS1L-MYB* by the presence of SNPs, transcription factor, and a therapeutic agent.

**Figure 3 diagnostics-12-01374-f003:**
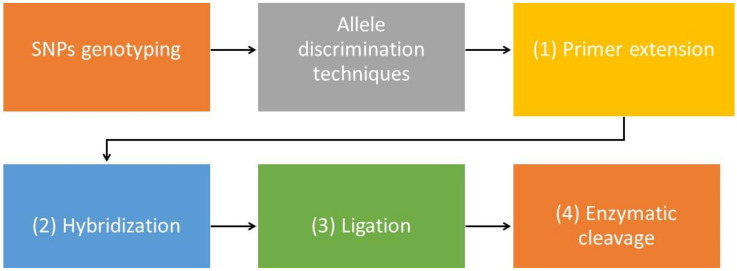
SNP genotyping in allele discrimination to increase the number of molecules for the detection of SNPs of interest in disease states.

**Table 1 diagnostics-12-01374-t001:** SNPs found in *XMN1-HBG2*, *BCL11A*, and *HBS1L-MYB* genes.

Locus	SNPs	Effects of SNPs	References
*XMN1-HBG2*	Rs782144	HbF level highly expressed, with T allele resulting in mild anemia and asymptomatic state in lenient disease.	[40,41,48]
		The high level of HbF may reduce pain crises, reduce risk of feto-maternal bleeding, reduce blood transfusion requirement, and unusual thalassemia.	[49,50]
*BCL11A*	Rs1018987	Rs1427407, rs10189857, and rs11886868 had no significant difference in <5% and >5% of HbF level.	[51]
	Rs4127407		
	Rs766432	The presence of T and G alleles in rs11886868 and rs4671393 is associated with amelioration of SCD phenotype through an increase in HbF level.	[52]
	Rs11886868		
	Rs4671393		
	Rs6729815	The allele C in rs11886868 and allele A in rs4671393 are linked to increased HbF levels (*p* = 0.026 and 0.028) but have no correlation with stroke.	[53]
	Rs1426407		
	Rs6545816		
	Rs7606173	The downregulation of *BCL11A* caused by rs11886868 results in high HbF production (11.36–49.40%) in the CC genotype, followed by CT and TT.	[54]
	Rs6545817		
		The presence of knockdown of *SOX6* and *KLF1* may reduce the silencing of the γ-globin gene and enhance HbF levels to more than 28.9%.	[55,56]
		Hydroxyurea treatment is associated with rs11886868, rs4671393, and rs1427407, which may ameliorate anemia.	[57,58]
		The HbF levels may be increased due to Rs11886868, which consists of the C allele; this also ameliorates the clinical phenotype.	[59]
*HBS1L-myb*	Rs4895441	Rs9399137 with C allele associated with increased HbF production and hematologic parameters.	[51]
	Rs66650371		
	Rs9399137	The decrease in *MYB* expression in rs66650371 is associated with high HbF synthesis to produce proper RBC and Hb levels.	[60,61]
	Rs11759553		
	Rs28384513		
	Rs9402686	The rs9402686 at sub-locus is associated with high HbF in the A allele.	[61]
		Minor allele frequency of rs9399137 or rs11759553 more than 5% may increase HbF levels and has been proposed as a potent target for therapeutic purposes.	[62,63]

## Data Availability

The data presented in this study are available on request from the corresponding author.

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
