# Peer review of "Single Nucleotide Polymorphisms in XMN1-HBG2, HBS1L-MYB, and BCL11A and Their Relation to High Fetal Hemoglobin Levels That Alleviate Anemia"

_diagnostics, 2022, doi:10.3390/diagnostics12061374_

Round 1
Reviewer 1 Report
This is a good manuscript trying to highlight the importance of 3 important polymorphisms that can increase the expression of fetal hemoglobin and ameliorate the signs and symptoms in anemic patients. However, before publishing the manuscript, I would like to request the authors to improve their manuscript.
- Please increase your effort to improve the manuscript (including English).
- Please justify the rationale, in terms of advantage and disadvantage of having the polymorphisms.
- Your Figure one is a Flow chart, it is not a Figure. Do you need to classify anemia for justifying the polymorphisms? If you need, then please classify anemia on the basis that relates to your paper's objective.
- Your Table 3.1 is basically Table 1. Please do not combine references.
- Include a separate section for describing how each of the polymorphisms regulates the expression of fetal hemoglobin. Please separate epidemiology and molecular aspects (regulation of HBG promoter).
- Add some illustration to the texts, so that the readers can understand what you are telling.
- Please make a section to describe (with illustration) how the polymorphisms can be detected (diagnosed) in patients and how that can improve clinical aspect / management of anemic patients.
- If you are describing only human genes, why do you indicate sometime genes in small letters? Are you referring to rodent experiments?
- Please make more clear description and increase content, as well as figures (illustrations).
Author Response
Dear reviewer,
Thank you for comments.
Please see the attachment.

Reviewer 2 Report
This review is very confusing and conflates genetic and acquired conditions. There are a lot of conceptual errors and faults in the interpretation of the literature cited. There is great confusion among genes, snps, qtl etc which are not appropriately used.
The whole text is unfocused. In the introduction the authors write about anemia of acquired as well as of genetic origin whereas the work describes findings relevant only to beta-hemoglobinopathies (genetics). The discussion focuses only on beta-hemoglobinpathies.
The authors should probably have focused the review only on beta-hemoglobinopathies and not on anemias in general.
On some cases the sentences are incomprehensible, at least for me, and the English is hostile.
Few examples:
Abstract,
“Anemia is caused by scarcity of red blood cells and low hemoglobin count, resulting in a lack of oxygen being transported to tissues and organs. “
Comment: Anemia is not cause by but is a condition were hemoglobin (Hb) concentration and/or red blood cell (RBC) numbers are lower than normal and insufficient for tphisiological needs
“Its severity has been observed to be less in patients when fetal hemoglobin (HbF) levels are augmented through the reactivation of the γ-globin gene. “
Comment: This is true only for beta-hemoglobinpathie.
Introduction
“Variations in HbF production in approximately more than 20 % of the world’s population is influenced by genetic modifiers in three genes, namely XMN1-HBG2, HBS1L-MYB and BCL11A [8,9] “
Comment: Rather, SNPs from these three loci together account for >20% of variation in HbF levels.
“Therapeutic agents (inducer drugs), such as hydroxyurea and azacytidine, are widely used in treating SCD, thalassemia and leukemia by elevating the HbF level of patients [19, 21-25 “
Comment: azacytidine is not used in SCD e beta thal. DNA methylransferase inhibitor 5-azacytidine was one of the chemotherapeutic agents used to reactivate HbF but it was quickly abandoned due to its toxicity and carcinogenicity.
Approximately 20 % to 50 % of polymorphisms in patients may be attributed to these genes [7].
Comment: Wrong again. The cited literature stated that 20-50% of HbF variance in patients with sickle cell anemia and healthy European Caucasians.
“The HBS1L-MYB is a protein-coding gene that modulates the level of HbF in erythroid progenitor cells, and is associated with severity of anemia, thalassemia and SCD [51]. “
Comment: The concept is wrong, and the bibliography is not cited properly.
The HBS1L-MYB is not a protein-coding gene but an intergenic region.
“The increase of HbF levels may be observed with downregulation of BCL11A with a zinc-finger cluster, which then allows the expression of γ-globin [61, 62]. The presence of C- terminal zinc finger cluster, downregulation of nucleosome remodeling and deactylase [NuRD], silence shifting of the 5’ δ-gene of BCL11A and de novo deletions (frameshift, nonsense, missense) will reduce the expression of BCL11A, thus enhancing HbF levels [61, 63]. “
Commet: This whole sentence is not understandable.
Table 1 does not follow the order of description of the text.
etc etc…
Author Response
Thanks for comments
Please see the attachment.

Reviewer 3 Report
It is well written review regarding to Single nucleotide polymorphisms in XMN1-HBG2, HBS1L-MYB and BCL11A and their relation to high fetal
hemoglobin levels that alleviate anemia.
Author Response
Thanks for the comments

Round 2
Reviewer 1 Report
The authors improved the manuscript significantly. Although there are rooms for further improvement, the current manuscript is in a good form.
Author Response
Dear reviewer,
Thanks for the comments.
The manuscript has been proofread by a profesional English editor
zefarina
Reviewer 2 Report
English should be revised
Author Response
Dear reviewer,
Thanks for the comments.
I resubmit the manuscript that has been proofread by a profesional English editor.
zefarina